# Effect of Irradiation on Reproduction of Female *Spodoptera litura* (Fabr.) (Lepidoptera: Noctuidae) in Relation to the Inherited Sterility Technique

**DOI:** 10.3390/insects13100898

**Published:** 2022-10-02

**Authors:** Madhumita Sengupta, Neha Vimal, Nilza Angmo, Rakesh Kumar Seth

**Affiliations:** Applied Entomology and Radiation Biology Lab, Department of Zoology, University of Delhi, Delhi 110007, India

**Keywords:** radio-genetic pest control tactics, Inherited Sterility Technique, *Spodoptera litura*, radio-sterilized female moth, sex pheromone, *PBAN*, *PBAN-R*, *Vg*

## Abstract

**Simple Summary:**

*Spodoptera litura* (Fabr.) is a serious Lepidopteran pest that can be controlled using the eco-friendly radio-genetic Inherited (F1) Sterility (IS) technique. In contrast with the conventional male-only release applied in IS techniques, this study was conducted to determine a suitable dose at which females can be fully radio-sterilized while leaving the more radio-resistant males partially sterile, meaning that both sexes could be irradiated together and released simultaneously in pest infested fields. Behavioral and molecular investigations ascertained 130 Gy to be a suitable dose for female sterilization, at which point their reproductive fitness in terms of calling ability and mating competence will not be significantly compromised. These irradiated females would be sufficiently viable to compete with wild females for mating with wild males, ultimately suppressing the pest population. The simultaneous release of sub-sterile male and sterile female moths using the IS technique might improve its efficacy, leading to more effective pest suppression.

**Abstract:**

Radiobiological investigations on the reproductive behavior of female *Spodoptera litura* (Fabr.) were conducted with the aim of determining the suitable radio-sterilizing dose for females in order to release them along with sub-sterile males for effective implementation of the Inherited Sterility technique against this pest. Calling and copulation duration significantly increased, while mating success, oviposition, fertility and longevity significantly decreased with increasing radiation dose (100–200 Gy) compared to control. In view of the effect of irradiation on mating behavior and reproductive viability of female *S. litura*, 130 Gy was identified as a suitable radio-sterilization dose. Further molecular studies were conducted to corroborate this dose for female sterilization, along with a higher dose of 200 Gy in order to validate the gradational response of ionizing radiation. GC-MS analysis indicated decreased sex pheromone titer at 130 Gy, which was more pronounced at 200 Gy. Pheromone-associated genes, *PBAN* and *PBAN-R* showed decreased expression at 130 Gy, and were drastically reduced at 200 Gy. The fertility-related *Vg* gene also showed a negative correlation with radiation exposure. Based on these radiation responses of female *S. litura*, 130 Gy might be considered a suitable dose for complete female sterility and its inclusion in sterile insect programs against *S. litura*.

## 1. Introduction

*Spodoptera litura* (Fabricius) [Lepidoptera, Noctuidae], is a serious polyphagous Lepidopteran pest, commonly distributed around south and south-east Asia and Oceania [1,2]. In India this pest is one of the major tropical pests feeding on a variety of crops like tobacco, tomato, cotton etc. This pest feeds on more than 40 botanical families cementing its status as a serious pest economically devastating the agricultural yield. It is a holometabolous insect and takes about a month to complete its life cycle consisting of egg, larva, pupa and adult stages of development. The larva is the major defoliator of the agricultural crop fields feeding largely on the leaves [2]. Devising a bio-rational control tactic for this lepidopteran pest was required as a result of development of immense insecticide resistance [3,4].

Autocidal pest control tactics such as the sterile insect technique (SIT) and the inherited sterility (IS) technique are eco-friendly and very efficient for integration in area-wide integrated pest management approaches [5]. Traditionally, the SIT/IS programme requires the operational target at a reasonable cost with sex-based separation of the male and female insects, the sterilization of the males by ionizing radiation and subsequent release of the radio-sterilized males in the target area. The sterilized insects are released at a sterile to wild male overflooding ratio high enough to enable the irradiated males to compete with the wild males to mate with the wild females. Depending on the dose administered, the irradiated males after mating with a wild female, produce no offspring (SIT) or sterile F1 offspring (IS) and this results in pest population suppression with each subsequent generation [6]. The SIT approach is considered a less efficient strategy for the management of lepidopteran pests, because of their high resistance to ionizing radiation which requires very high doses of gamma radiation to completely sterilize the male insects. These high doses would compromise the biological quality of the sterile moths as compared with the wild ones due to the resulting somatic damage [7]. 

In all insects, exposure to ionizing radiation causes dominant lethal mutations (DLMs) in the chromosomes, but in Lepidoptera, these appear mostly during the later stages of the embryonic development [8]. The gamma radiation induced fragmentation of chromosomes is however, transferred through the germ cells to the F1 progeny inducing sterility [9]. Therefore, in lepidopteran males (unlike dipterans), multiple chromosomal aberrations can only be obtained using high doses of radiation leading to DLMs effective enough to cause sterility [6]. This is the reason high doses of radiation doses (350–500 Gy) are required to completely sterilize lepidopteran males [10]. 

In the F1 sterility technique, a sub-sterilizing gamma dose is used for the males that are destined for release for lepidopteran pest suppression [11]. Consequently, the Lepidoptera oriented radiation studies were conducted on dose evaluation, radiation biology, development of radio-resistance, mass rearing optimization, mating competence of irradiated moths and simulation modelling to ultimately increase the efficacy of the radio-genetic techniques. Several studies highlighted the great potential of SIT/IS for the suppression of several lepidopteran pests [7,12,13]. 

The germ cells of lepidopteran females are more sensitive to irradiation than the sperm of males [14]. Therefore, the range of 100–200 Gy is in most cases enough to completely sterilize female Lepidoptera [4,6]. Therefore, in IS technique using release of both sexes, a common irradiation dose is required to completely sterilize the females and partially sterilize the male moths. Therefore, in IS technique pertaining to simultaneous irradiation and combined release, a common irradiation dose was required to completely sterilize the females and partially sterilize the male moths.

It is not known whether the release of sterile females together with sterile males would bring any benefit to SIT/IS programme against lepidopteran pests. Combined release of irradiated male and female moths using single ionizing dose may be convenient as it does not require the laborious process of sex-based pupal segregation [15]. Specific to this pest, *S. litura,* the previous studies have shown that a range of 130–150 Gy of ionizing radiation was deemed suitable for partially sterilizing the males of this species, which when crossed with normal females lead to sterile F1 progeny [16]. 

The mating competence and reproductive fitness of the moths could be associated with their pre- and post-mating behavior. The pheromone production and release are important for the calling efficacy by the female moths, which may govern the mating performance of these irradiated females. 

The aim of this study was to identify a suitable gamma radiation dose to induce complete sterility in female moths without majorly affecting their reproductive fitness that was studied in terms of various reproductive parameters. 

The effect of radiosterilizing dose was also evaluated on the pheromone titers and the expression of important reproductive genes such as the pheromone biosynthesis activating neuropeptide (*PBAN*), PBAN- receptor (*PBAN-R*) and the vitellogenin *(Vg*) gene, in order to correlate these molecular attributes to the sterile female’s performance. 

## 2. Materials and Methods

### 2.1. Culture and Maintenance of Spodoptera litura

The colony of *S. litura* was maintained in BOD incubators (P.L. Scientific, New Delhi, India) under the following environmental conditions: a temperature of 27 ± 1 °C, relative humidity of 70 ± 5% and a photoperiod regime of 12 h light: 12 h dark.

Four pairs of adult moths were placed in each Perspex/nylon cage (V.L Enterprises, New Delhi, India) for the purpose of mating. Cages of different sizes were used as required for the experiments. A cage of 20 × 20 × 20 cm was used for the experiment involving 4–6 pairs and a cage of 60 × 60 × 60 cm was used for 10–15 pairs. Cotton swabs saturated with 10% (*w/v*) honey/water solution were placed in small plastic containers (6 × 2 × 2 cm^3^, All Serve scientific, New Delhi, India) and kept in the adult cages for feeding. The cotton swabs were replaced every day. One leaf of a castor oil plant (*Ricinus communis* L.) was put in each cage held in a 100 mL Duran bottle (DWK life sciences, Mainz, Germany) containing water with the stalk of the leaf dipped in it to serve as an oviposition substrate. The oviposition trap was changed daily after egg collection. The eggs were collected from the containers with brushes, surface sterilized with 0.01% sodium hypochlorite solution for 2 min and kept for incubation in the BOD incubators under the following environmental conditions: a temperature of 27 ± 1 °C, relative humidity of 80 ± 5%, and a photoperiod regime of 12 h light: 12 h dark. The larvae were reared in the same BODs on a semi-synthetic diet [17], and the adults emerging from the pupae were separated from culture and kept in 1 L glass jars (Yera glassware, Vadodara, Gujrat, India) until eclosion and then the adults were placed in the cages for mating. 

### 2.2. Irradiation Treatment

Females, 0–1 days old, were exposed to a range of radiation doses (100, 130, 150 and 200 Gy) in a Co^60^ research irradiator, Gamma Chamber 5000 (Board of radiation and isotope technology, Navi Mumbai, India) at dose rate of 0.625–0.429 KGy/h. The irradiator was located at the Institute of Nuclear Medicine and Allied Sciences (INMAS) of the Defense Research and Development Organization (DRDO) in Delhi-110054. Non-irradiated female moths served as the control group and were kept under the same conditions (light, temperature, etc.) described in Section 2.1. 

### 2.3. Behavioral Study

To study the pre-mating behavioral responses of the females, the newly emerged adults were placed in cohorts of four pairs in a cage of 20 × 20 × 20 cm. Parameters such as female calling (the time between the female lifting up the abdomen and induce pulsating movement in the lower abdomen to release pheromones and the initiation of mating) and the duration of mating or copulation duration (i.e., the period between the insertion of male genitalia into the female and retraction of the male genitalia from the female abdomen) were recorded. Each cohort of four pairs constituted one replicate, and this study contained a total of 15 replicates.

The mating success was determined by observing the number of females with spermatophore(s) in their *bursa copulatrix* (indicative of a successful mating) in a cohort of 12–15 adult pairs kept in a cage of 60 × 60 × 60 cm. The females from each cage were dissected for their bursa after their death, and spermatophores were counted for each female. The percentage of the total number of females with spermatophore(s) out of total number of females observed was computed to calculate average mating success for that cohort. Each cohort of 12–15 pairs constituted one replicate, and this study contained a total of 15 replicates.

For the oviposition experiment, each cohort of four pairs constituted one replicate, with a total of 15 replicates in total. For each replicate, the eggs were collected every day during the ovipositional period, and an average of total egg laying per mated female was computed as an ovipositional response.

For egg fertility, the average egg hatch of 10 egg samples (with each sample containing 50–100 eggs) from each cohort of 3–4 pairs constituted one replicate (*n* = 15). The egg samples were randomly collected from the oviposited eggs during the first 5 days of egg laying in each regimen.

The survival and mortality of the moths were recorded daily until the moths died in the cage. The average longevity of the moths in a cohort of 4 pairs per cage constituted one replicate and 15 such replicates were evaluated for assessing the longevity of the moths in a specific regimen. 

### 2.4. GC-MS Analysis

After emergence, the virgin adult females were collected at dusk and their pheromone glands dissected 3–4 h after lights off in the scotophase. The pheromone glands from 10 females were used as a replicate for gas chromatography mass spectrometry (GC-MS), and three replicates were used for each treatment. The pheromone glands (PGs) from 10 adult females were dissected out, and all 10 PGs of each replicate were pooled in a GC glass vial (2 mL vials, Tarsons, Kolkata, West Bengal, India) by soaking them in 1 mL *n*-hexane with tridecyl acetate as an internal standard (Sigma Aldrich, St. Louis, MO, USA) for half an hour at room temperature to extract the pheromone components. The pheromone components were then purified by sonication followed by ultrafiltration with a 0.2-micron syringe filter into another 2 mL GC glass vial. These vials containing the pheromone extracts were stored at −20 °C until further analysis. The pheromone extracts were then analyzed by GC-MS in an Agilent Technologies 6890 N gas chromatograph with an HP-5 column and a flame ionization detector following the operational protocol specified by Lu [18]. The pheromone components were identified by comparing their retention times and mass spectra with the retention time of the internal standard (tridecyl acetate) used in the mass spectra databases. The pheromone components were functionally categorized with the help of information from the literature and from additional MS databases. The major pheromone components were quantified based on the peak areas of the components with help of known concentration of internal standard at the time of GC-MS analysis. Kovat’s retention indices were used with the mass spectroscope to convert individual retention times into constants and identified as per the components with these known constants already present in the databases. GC-MS analysis was performed for all samples treated with different radiation doses, and the major pheromone components were selected as the components that had the highest titer and that made up most of the pheromone mixture composition, i.e., eicosyl acetate (tricosyl acetate), tetracontane, tetradecadienyl acetate and 2,6,10,15-tetramethyl heptadecane. These components were then subsequently plotted according to their concentration in the pheromone mixture from each sample and each irradiation treatment. 

### 2.5. Sampling, RNA Isolation and cDNA Synthesis

For gene expression analysis, the freshly eclosed females were dissected according to the required tissue. The head and upper thorax (consisting of the brain + sub-esophageal ganglion (SOG)) of the adult female moths were excised to assess the gene expression of the pheromone biosynthesis activating neuropeptide (*PBAN*). The pheromone glands of adult females were sampled for the study of the PBAN receptor (*PBAN-R*) and the fat bodies of adult females [19,20,21] were used for the study on the vitellogenin (*Vg*) expression. The brain + SOG, pheromone glands and fat body samples were collected during the scotophase. Each sample consisted of a single tissue from one moth, and a total of 6 replicates were used for each specific regimen.

The total RNA was isolated from the above tissues using the Trizol reagent followed by the phenol-chloroform extraction method and treated with DNase I [22]. The RNA concentration in each sample was measured using the NanoDrop 2000 C (Thermo Fischer Scientific, Waltham, MA, USA). The 260/280 ratio for absorbance between 1.8 to 2.0 was considered pure for the RNA samples. A 1 μg RNA aliquot treated with DNase was used for single stranded cDNA synthesis by Revert Aid First Strand cDNA synthesis kit (Thermo Fischer Scientific, Waltham, MA, USA).

### 2.6. Measurement of Gene Expression through qPCR

Gene-specific direct primers were developed and used for qPCR (quantitative real-time PCR) (Table 1). The quality and specificity of each primer was tested with the respective tissues by performing RT-PCR on the primers with the tissues and the primer pair showing a single peak in the melt curve was chosen. qPCR was performed using SYBR green (Thermo Fischer Scientific, Waltham, MA, USA) as intercalating dye for each sample and qPCR was conducted on Applied Biosystems ViiA7 real-time PCR system. In total, 40 cycles were run, with each cycle being 75 s long and having a melting period of 15 s at a temperature of 95°C and an annealing period of 60 s at a temperature of 60°C. *EF1* was taken as the reference gene for normalization. The reference and gene of interest were run in duplicates for all treatment groups, with each treatment group containing 6 replicates (*n* = 6), in a single plate to avoid variations. Each 384-well PCR plate (Applied Biosystems, Thermo Fischer Scientific, Waltham, MA, USA) included the non-template and sample controls without reverse transcriptase used in cDNA synthesis, and an endogenous control of *EF1* gene (housekeeping gene). The relative expression of the targeted gene was determined using the ΔΔCt method [23]. This method uses the difference in cycle threshold (Ct) value between reference and target genes to calculate an estimated fold change in the target gene. In this method, firstly, ΔCt value was calculated by subtracting Ct of the target gene from reference gene and then, ΔCt was normalized against ΔCt of the calibrator sample (pool), which consisted of cDNA mix from all the groups compared. The negative of the resultant ΔΔCt value powered to 2 (2^−ΔΔct^) was plotted as the relative mRNA expression of the target gene.

### 2.7. Statistical Analyses

All statistical analyses were performed using Graph Pad Prism software (version 9.0, San Diego, CA, USA), with 6–15 replicates as specified in the text. One-way analysis of variance was used, followed by Tukey’s post hoc test for multiple comparisons of the behavioral parameters and the different radiation treatments. Similarly, one-way ANOVA followed by Tukey’s test was used for GC-MS analysis and the gene expression profiles to test the significant difference at the *p* < 0.05 level of the different irradiation treatments of female moths compared with the control. Simple linear regression analysis was performed to compare the dose response of behavioral parameters of irradiated female moths for which the significance was taken at R^2^ > 0.8.

## 3. Results

### 3.1. Radiation Effects on Mating Behavior and Reproduction

The calling of female moths of *S. litura* usually started 10–15 min after the onset of the scotophase. The females exhibited peak calling activity up to 3 days after emergence. Females that were irradiated called for significantly longer, and the initiation of mating was delayed as compared with untreated control females. Figure 1a showed the effect of gamma doses (0–200 Gy) on the calling duration of female moths. The average duration of calling of the untreated control females was 29.6 min, whereas 200 Gy irradiated females called on average for 58.5 min (in the presence of untreated males). The duration of calling increased with increasing radiation dose. One-way ANOVA analysis indicated that irradiated females called for significantly longer compared with untreated females (F *_(4,70)_* = 5.83, *p* < 0.05, R^2^ = 0.81), but there was no significant difference in duration of calling between the irradiated females (130–200 Gy).

Figure 1b shows the positive correlation between copulation duration and increasing radiation doses (ANOVA; F *_(4,70)_* = 4.33, *p* < 0.05, R^2^ = 0.97). The average copulation duration of untreated control females was 54.2 min, whereas 200 Gy irradiated females were found in copula for an average time of 84.4 min (paired with untreated males). The increase in copulation duration as compared with the untreated control females was significantly longer for females irradiated with 130 Gy and above.

The mating success of the untreated control females was 93.6%, whereas only 79.8% of the 200 Gy irradiated females mated with untreated males (Figure 1c). There was a significant negative correlation (R^2^ = 0.92) between mating success and increasing radiation dose, and the mating success significantly decreased with increasing radiation dose as compared with the untreated control (ANOVA; *p* < 0.05, F *_(4,70)_* = 6.49) (Figure 1c).

Untreated females oviposited on average 1728 eggs, whereas 200 Gy treated females mated with untreated males oviposited ~680 eggs as shown in Figure 1d (59.8% decrease). The number of eggs oviposited declined gradually with increased radiation doses with a strong correlation coefficient of 0.91 (ANOVA; *p* < 0.05, F *_(4,70)_* = 28.9). 

The average egg fertility (egg hatch) of the control females was 91.6%, whereas no eggs hatched that were oviposited by 200 Gy irradiated females that had mated with untreated males. The egg hatch decreased significantly with increasing dose (ANOVA; *p* < 0.05, F *_(4,70)_* = 15, 213, R^2^ = 0.76) (Figure 1e). The fertility of 100 Gy irradiated females was only 2–5% and nil at 130 Gy and above.

The untreated control females lived on average 9.2 days, whereas the mean longevity decreased to 4.2 days for 200 Gy irradiated females (Figure 1f). The longevity of irradiated females was thus reduced by 22.8–32.4% for irradiation doses ranging between 100 Gy and 150 Gy and 54.6% for 200 Gy treated females (ANOVA, *p* < 0.05, F *_(4,70)_* = 24.45). The linear regression coefficient between mean longevity and radiation dose was highly significant (R^2^ = 0.98).

### 3.2. GC-MS Analysis of Pheromone Profiles

The GC-MS analysis of the pheromone gland extracts of the females showed different pheromonal components released by the moths at the onset of the scotophase. The pheromone components are functionally annotated and summarized in Table 2 [24,25,26,27,28,29,30,31,32,33,34,35,36,37,38,39,40,41,42,43,44,45]. The pheromone components were classified into the following categories: (1) sex attractants, (2) sex pheromone precursors, (3) orientation pheromones, (4) ovipositor extensor/releasor, and (5) copulation releasor/terminator. 

To assess the effect of irradiation on the titers of these components, the sex attractants that had the highest titer and that made up most of the pheromone mixture composition were selected, i.e., eicosyl acetate (tricosyl acetate), tetracontane, (Z,E)-9,12-tetradecadienyl acetate and 2,6,10,15-tetramethyl heptadecane.

The female moths irradiated with 130 Gy and 200 Gy showed a significant decrease in their sex attractant concentration. For instance, the eicosyl acetate titer decreased to ~8 ng/uL (decreased by 28.6%) in 130 Gy and to a mere 2.8 ng/uL (decreased by 35.7%) in 200 Gy irradiated females compared with the titers in the untreated control females, which corresponded to ~12 ng/uL. Similarly, the titers of tetracontane and tetradecadienyl acetate decreased to 6 ng/uL and 23 ng/uL (decreased by ca. 40%) in 130 Gy and to 3.5 ng/uL and 12.5 ng/uL in 200 Gy irradiated females (decreased by ca. 70%), respectively, compared to the control females, where tetracontane was 16 ng/uL and tetradecadienyl acetate was 28 ng/uL in the pheromone mix. The titers of the copulation terminating pheromone component 2,6,10,15-tetramethyl heptadecane decreased to 6.2 ng/uL (decreased by 15.4%) and 4.6 ng/uL (decreased by 40%) in 130 Gy and 200 Gy irradiated females, respectively, compared with the 15 ng/uL present in the untreated control females (Figure 2).

### 3.3. Irradiation Effects on Reproductive Genes

#### 3.3.1. *PBAN* Expression Levels

Expression levels of the *PBAN* gene in 130 and 200 Gy irradiated females were significantly decreased compared with the untreated control females (ANOVA, F *_(2,15)_* = 101.3, *p* < 0.05) (Figure 3a).

#### 3.3.2. *PBAN-R* Expression Levels

Expression levels of the *PBAN-R* were significantly reduced in 130 and 200 Gy irradiated females compared with the untreated controls (ANOVA: F *_(2,15)_* = 42.2, *p* < 0.05) (Figure 3b). This is in agreement with our findings on the decreased pheromone production in the pheromone gland extracts of the irradiated females (Figure 2). 

#### 3.3.3. *Vg* Expression Levels

Expression levels of the vitellogenin (*Vg*) gene were significantly reduced in 130 Gy and 200 Gy irradiated females compared with the untreated control females, and might be noted as being a pivotal factor for the induction of sterility in irradiated females (ANOVA: F *_(2,15)_* = 4.6, *p* < 0.05) (Figure 3c).

## 4. Discussion

In SIT/IS programs against Lepidoptera, male and female moths are usually released simultaneously, because genetic sexing strains that would allow the release of males only are not available. In addition, it obviates the need for the labor-intensive procedure of sex-based separation of the sexes by hand [46]. However, the release program should also provide irradiated moths that are competitive with native moths in order to induce the required level of sterility in the wild population [17]. As part of the research to assess whether the release of sterile female Lepidoptera together with sterile males can result in additional benefits to the program, it was deemed useful to understand the effect of female irradiation on the mating and reproduction of the moths, using *S. litura* as a study organism.

In view of the high intrinsic radio-resistance of Lepidopteran species and the different radio-sensitivity of the sexes [10], a common radiation dose has to be selected at which the females would be fully sterilized while the males would be partially sterilized. Hence this study was carried out to assess a series of behavioral, physiological, and molecular responses of female moths to radiation. This study provides an insight into the behavioral, physiological, and molecular responses exhibited by female *S. litura* when exposed to sterilizing and higher sub-lethal radiation doses. The physiological and molecular changes in female moths in response to ionizing radiation were previously unknown in Lepidopteran pests, including *S. litura*.

Various reproductive parameters of females, including calling duration, copulation period, mating success, oviposition, fertility, and longevity, were studied in response to radiation doses ranging from 100 to 200 Gy. The irradiation treatment resulted in a longer duration of calling and a longer premating period compared to those of untreated females, which indicates that the irradiation treatment reduced the potential of females to attract male moths. This might be related to a decrease in the pheromone production of irradiated females. Similarly, the copulation duration of irradiated females when paired with normal males was significantly longer, and this might negatively affect the reproductive competence of the female moths, as their re-mating incidences would be limited due to the added energy expenditure [47].

In previous studies on moths like *Eldana saccharina*, it was found that at lower doses of irradiation, control insects did not discriminate between untreated and treated moths for the purpose of mating, thereby exhibiting uncompromised mating percentage. Irradiated moths also exhibited a comparable mating percentage, which is in accordance with the mating success obtained in the results [48,49]. The mating success of the irradiated females was comparable with that of untreated females in the dose range of 100–150 Gy, but mating success was significantly reduced at 200 Gy, which might ultimately indicate compromised reproductive quality. Mating success is one of the crucial criteria for determining the mating competitiveness of treated insects in the field.

There was a steep decline in oviposition with increasing radiation dose. The criterion of fertility was important in selecting a suitable radiation dose that would be able to provide complete sterilization to female moths so that they could efficiently contribute towards the containment of the pest population, while also retaining sufficient reproductive prowess to be able to compete with the wild population. At 130 Gy, female reproductive behavior was not drastically affected, although it was associated with negligible fertility in the treated females. The effect of irradiation on the fertility of female moths has been reported in the previous studies, too [50,51].

The longevity of 130 Gy treated female moths was reduced by 23% compared to untreated females, whereas a treatment of 200 Gy reduced their longevity by more than 50%. Keeping in view longevity, premating behavior and mating success, a dose of 130 Gy would be an appropriate dose to use in a sterile insect release program [52]. These radio-sterile female moths (at 130 Gy) would act as sperm sinks by mating with normal (unirradiated) wild males that would in turn add to the efficiency of released partially sterilized male moths (also at 130 Gy) in the IS program. The radiation dose of 130 Gy induced sterility in females, but reasonable mating competence was retained in terms of female calling, mating duration, mating success, oviposition, and longevity.

The applicability of a combined irradiation and release program is primarily based on the complete sterilization of female moths, which would therefore act as sperm sinks for wild males present in the field. Irradiated sterile females that retain their reproductive competence on par with unirradiated female moths would also help in mating disruption of wild insects by dispersing their pheromones in pest-infested fields. The pheromone titer results in this study suggest that at a dose of 130 Gy, the female moths are able to produce sufficient amounts of pheromones to be able to attract wild males. These findings indicate that combining the release of radio-sterilized females with sub-sterile males would further benefit IS programs. These observations were further validated by the molecular studies performed on the irradiated female moths.

Insufficient pheromone production might have been a factor resulting in prolonged calling and copulation periods. It has been reported in previous studies conducted on different moths that irradiation, especially at high doses, might adversely affect the pheromone production, secretion and intraspecific communication between moths [53]. The titers of major pheromonal components in female moths irradiated at doses of 130 Gy and 200 Gy were compared with those in untreated control moths. There was an evident decline in the quantity of sex attractants produced in the pheromone glands due to the irradiation of the female moths. The decline in titer was significant in the 130 Gy irradiated females, but the decrease was drastic in the 200 Gy treatment group. This indicated that the female calling capacity was not as markedly affected in the 130 Gy treated females as in the 200 Gy treated females. The copulation releasing pheromonal component was also decreased with increase in radiation dose which was in consonance with similar impact on the mating and copulation periods in *S. litura*. These results further reaffirmed that 130 Gy might be used as a suitable radio-sterilizing dose for females, as they were still able to produce a sufficient amount of pheromone components to be able to attract males at this dose. A dose higher than 130 Gy might negatively impact the calling efficiency of the irradiated females and affect their chances of mating with wild males.

The pheromone biosynthesis activating neuropeptide (PBAN) is produced by the neurosecretory cells (NSC) of the sub-oesophageal ganglion (SOG) in insects, and is responsible for the activation of pheromone biosynthesis [54]. The reduced pheromone production and calling efficiency at the higher radiation dose (200 Gy) implied that irradiation might have an impact on this gene’s expression. In the present study, *PBAN* gene expression was significantly decreased in 130 Gy treated females compared to the untreated control females. This decrease was even more pronounced in the 200 Gy treated females. This observation could be correlated with the insufficient pheromone production observed in the PGs of irradiated females in GC-MS analysis. This decrease is also in agreement with our previous study on the effect of radiation doses on relative expression and rhythmicity of the *PBAN* gene in *S. litura*, where gamma radiation, despite resulting in a decrease in gene expression, did not affect the rhythmic expression of this gene [55]. Therefore, this further validates that at a dose of 130 Gy, the female moths are still capable of exhibiting precopulatory behavior comparable to that of untreated females.

The PBAN released from the SOG goes to the corpora cardiaca (CC) and is subsequently transported through the hemolymph to the pheromone glands, where it is attached to the PBAN receptors (PBAN-R). As a result, pheromone production is activated in the pheromone glands [56]. In this context, the expression of the PBAN receptor (*PBAN-R*) gene was also studied under the influence of radiation. The relative expression of the *PBAN-R* gene was reduced in 130 Gy irradiated females compared to in untreated control females, and this decrease was even more pronounced in the 200 Gy treated females. This further indicated that a low-range dose of 130 Gy did not considerably compromise the calling efficiency of female moths.

Vitellogenin is an important egg yolk protein responsible for the development of eggs [57]. Hence, vitellogenin is considered a crucial factor towards female’s reproductive viability. Therefore, the female radiation exposure was also evaluated on the vitellogenin (*Vg*) gene expression in the present study. Vitellogenin gene expression also showed a significant decline in 130 Gy and 200 Gy irradiated females, which could be vital for inducing sterility at these doses. This finding is in agreement with those reported in previous studies on *Cadra cautella*, *Musca domestica*, etc., suggesting that the impact of radiation was similar on the expression of vitellogenin and egg viability [58,59]. The considerably lower relative expression of this gene at 130 Gy further validated this gamma dose as a suitable sterilizing dose for female *S. litura*.

## 5. Conclusions

Based on these findings on the reproductive parameters of irradiated females paired with untreated male moths, a gamma dose of 130 Gy might be selected for the radio-sterilization of female moths of *S. litura*, showing 100% sterility without completely compromising their reproductive fitness and competence. This gamma dose (130 Gy) also conforms with the selection of a sub-sterilizing gamma dose to be employed for males when performing the F1 sterility technique against this pest moth [17,60]. Therefore, a radiation dose of 130 Gy might be considered as an appropriate dose for irradiating both sexes at the same time without sex separation towards simultaneous release of fully sterilized female moths along with sub-sterilized male moths when employing the IS technique. Furthermore, mating competitiveness studies are in progress pertaining to the simultaneous release of fully sterile female moths and sub-sterile male moths of *S. litura* in order to establish this management modality in terms of practical implementation.

## Figures and Tables

**Figure 1 insects-13-00898-f001:**
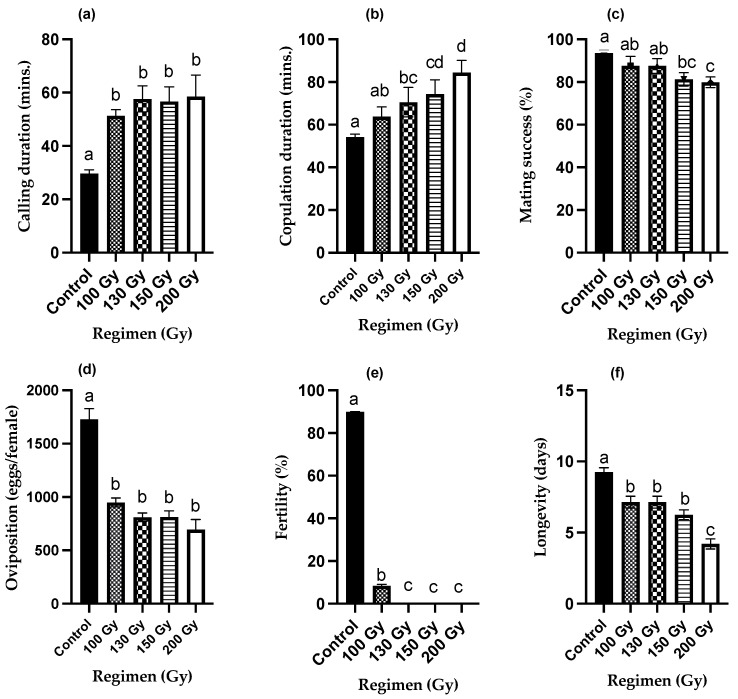
Effect of ionizing radiation on reproductive behavior of female *Spodoptera litura*. (**a**) Calling duration (min), (**b**) copulation duration (min), (**c**) mating success (%), (**d**) oviposition (eggs/female), (**e**) fertility (%), (**f**) longevity (days). *n* = 15, percentage (%) data was arcsine transformed before ANOVA, whereas the data representing the bars are back transformations. Means ± SE followed by same letters are not statistically different at *p* < 0.05 (one-way ANOVA followed by Tukey’s multiple comparison test).

**Figure 2 insects-13-00898-f002:**
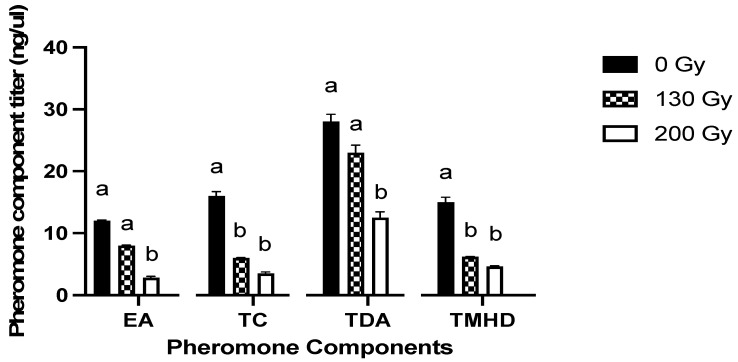
Effect of radiation on titers of major pheromone components of female *Spodoptera litura*. EA-Eicosyl acetate, TC-Tetracontane, TDA-(Z,E)-(9,12)-Tetradecadienyl acetate, TMHD-2,6,10,15-Tetramethyl heptadecane. Means ± SE followed by the same letters within each pheromonal component are not statistically different at *p* < 0.05 (one-way ANOVA followed by Tukey’s multiple comparison test). Average reading in a sample pool of 10 adult female pheromone glands (PGs) constituted one replicate (*n* = 3).

**Figure 3 insects-13-00898-f003:**
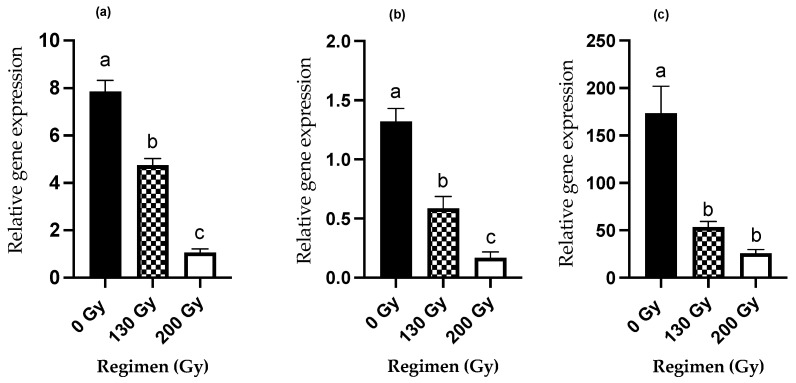
Expression profiles of reproductive genes in female *Spodoptera litura* irradiated at different gamma doses. (**a**) *PBAN* gene expression, (**b**) *PBAN-R* gene expression, (**c**) *Vg* gene expression. Means ± SE followed by the same letters are not statistically different at *p* < 0.05 (one-way ANOVA followed by Tukey’s multiple comparisons test).

**Table 1 insects-13-00898-t001:** Details of primer sequences used for the quantitative real time PCR (qPCR).

S.No.	Gene	Primer Sequences	Accession No.
1	*EF1 α*	F 5′-GACAAACGTACCCATCGAGAAG-3′ R 5′-GATACCAGCCTCGAACTCAC-3′	XM_022965580.1
2	*PBAN*	F 5′-CTCGGCAGGACGATGAATTT-3′ R 5′-CTGTTGGTACTCCTGACCATTC-3′	KP_006328.1
3	*PBAN-R*	F 5′-GTATTCTTCGTGGTGCCTATGT-3′ R 5′-CGAGAGCTTCTTCACTGGATG-3′	KM_023791.1
4	*Vg*	F 5′-GTTGTCTGCCGGTCGAATAA-3′ R 5′-GACTTTCCTGAGTCTGTGTGAG-3′	EU_095334.1

**Table 2 insects-13-00898-t002:** Functional categorization of different pheromone components obtained from GC-MS analysis of pheromone gland extracts of female *Spodoptera litura*.

Functional Category	Pheromone Components	Reference
Sex attractants	1-tetradecyl acetate	[24]
	11-oxohexadecanoic acid	[25]
	(Z,E)-9,12-Tetradecadienyl acetate	[26]
	1-docosanol acetate	[27]
	Bis(2-ethylexyl) phthalate	[28]
	Tetracontane	[29]
	(E,E,E,E)-squalene	[30]
	Hexatriacontane	[31]
	2-methyl octacosane	[32]
	Tricosyl acetate	[33]
Sex pheromone precursor	*n*-Hexadecanoic acid	[34]
	9,12-octadecadienoic acid	[35]
	9-octadecanoic-(Z)-methyl ester	[36]
	*n*-Eicosane	[37]
	2-methyl octasane	[38]
	(9,12)-hexadecadieonoic acetate	[39]
	(Z)-7-hexadecanal	[40]
	1-heptadecane carboxylic acid	[41]
Orientation pheromone	*n*-docosane	[42]
Ovipositor releasor	1,2-benzenedicarboxylic acid	[43]
	Heneicosane	[44]
Copulation releasor/terminator	2,6,10,15-tetramethyl heptadecane	[45]

## Data Availability

All data are contained within the article.

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
