# Peer review of "Effect of Irradiation on Reproduction of Female Spodoptera litura (Fabr.) (Lepidoptera: Noctuidae) in Relation to the Inherited Sterility Technique"

_insects, 2022, doi:10.3390/insects13100898_

Round 1

Reviewer 1 Report

Title of the study: “Effect of irradiation on reproduction of female Spodoptera litura (Fabr.) (Lepidoptera: Noctuidae) in relation to the Inherited Sterility technique“

This study investigates the irradiation dose inducing complete sterility in females and partial sterility in males Spodoptera litura (Fabr.). In a series of experiments, the authors demonstrated that 130 Gy is a suitable dose for sterilizing females without significantly compromise their reproductive fitness. Moreover, the sex pheromone extracts and pheromone‘s associated genes were also investigated assessing the effects of irradiation exposure. The study is comprehensive and well-written providing important fundings for the development of Inherited Sterility technique against this serious Lepidopteran pest, therefore, interesting, and worth publishing.

Please find below some minor suggestions to improve the manuscript.

Lines 82-85. Please erase the sentence and join the citation with the previous period as: “The germ cells of lepidopteran females are more sensitive to irradiation than the sperm of males [14].”.

Lines 86-88. Please erase the sentence and citation [15]. This information is not clearly explained and superfluous.

Lines 88-92. Please modify accordingly: “Therefore, in IS technique using both sexes releases, a common irradiation dose is required to completely sterilize the females and partially sterilize the male moths”.

Line 95. Missing citation.

Lines 101-103 Repetitive, already explained above. Please erase “…, in view of the higher radio-sensitivity of female moths than male ones. Hence this common gamma dose would avoid the sexing process and separation of sexes.

Lines104-105. Please rephase this sentence and merge it together with the previous once stating the aims of the study.

Lines 105-109. Please try to develop more this part. Here, you already explained the aims, but you did not introduce before the reasons of evaluating pheromones and genes expressions. Please introduce few lines regard this matter before starting to list the study aims.

Line 115: or hours or h.

Line 122: Ricinus communis L.

Lines 127-128. Remove: “… (P.L. Scientific, New Delhi, India)…“.

Lines 161-164: Can you please explain how and when the eggs for the oviposition and fertility experiments were collected?

Lines 165-166. How was the adult longevity calculating? Is it the average life spam of the first and the last adults died in each cohort? Please explain.

Line 200. Please explain where those adult female moths came from. Are the same used on the GC-MS analysis? And how many females were used?

Line 251. Please substitute “10-15 m” with “10-15 min. “. Do it in the whole chapter.

Line 255. Please rephrase: “… duration of calling and the irradiation dose is summarized….”.

Line 261: Why only two doses are here specified in brackets (130-200 Gy)? Please remove the brackets if the sentence is true for all irradiation doses.

Line 263: Remove one bracket.

Line 279. Maybe better to say significantly instead of drastically. I see now this term even above. Drastically does not provide an information about the significance of the results. Please rephrase when possible.

Line 286-287. Correct ‘’…was significant (R2=0.98).’’.

Line 291. Modify as: “...Longevity (days); n=15. Percentages (%) data were arcsine transformed before…”.

Line 315. Move ‘’ n=3’’ at the end of the sentence.

Discussion paragraph. Along the entire paragraph there is a nice summary of the purposes and results of this study, but rather little is discussed. Please try to be less repetitive and add more information regards previous works. Particularly try to discuss previous founding in literature regards Lepidoptera and other species used in IS/SIT programs and put in the contest of this study.

Please be aware when citing chapters from the SIT book of its latest: Dyck, Victor A., Jorge Hendrichs, and Alan S. Robinson. Sterile insect technique: principles and practice in area-wide integrated pest management. Taylor & Francis, 2021.

Reviewer 2 Report

Dear Authors

I have read this paper several times, and this is the case, I can not comment the paper in negative way. I consider one of high interest, I would be happy to see this paper published as soon as possible.

Thank you

Reviewer 3 Report

The manuscript by Sengupta et al describes the impacts of a 130 Gy radiation dose on females of Spodoptera litura. The authors demonstrate that this dose of radiation not only renders the females functionally sterile, but also changes their mating pheromone production, and reduces expression of genes associated with female reproductive physiology (vitellogenin) and mating behaviors (PBAN and PBAN-R). The authors argue that this dose of radiation may help a SIT program, where it is not practical to sex-sort the insects before release, as the co-release of these irradiated females with partially-sterilized males at this dose could be an effective population control technique.

The manuscript is well written, the data are clearly presented, and the results are accurately interpreted. The impacts of radiation on the females are clear, although it is not clear whether these changes would be of any particular use in a SIT/IS strategy. I found the experimental results interesting, but regret that I did not find the rationale for this moderate radiation dose compelling. The final statement of the manuscript states the real deficiency of this study: “Further, the mating competitiveness studies are required pertaining to simultaneous release of fully sterile female moths and sub-sterile male moths of S. litura to establish this management modality in terms of practical implementation.” This single sentence is the only place in the manuscript where the authors note a lack of information on the males, and I believe that much more needs to be said about this strategy in terms of impacts on males. The following are some points that should be addressed:

1.       In Figure 1, untreated males were paired with untreated females, while treated females were paired only with treated males. For all tests, treated and untreated males should have been paired with treated and untreated females, in all combinations.

2.       It was noted in the Introduction that high doses of radiation are required to sterilize male moths, but such doses can weaken the males and render them less competitive. What impacts are observed on males subjected to the 130 Gy treatments?

a)       Do irradiated males respond effectively to calling females? How effectively can they fly to find females?

b)      It is indicated males at this treatment dose are partially sterile, but more information is needed. Is their mating success reduced, and how many fewer progeny do they produce? Are all their progeny sterile?

c)       Vg transcripts levels were lower in irradiated females; could some seminal protein transcripts be similarly impacted in males?

d)      Given that this moderate radiation dose had impacts on non-gonadal tissues in females (e.g. head tissues for pheromone production), one would expect similar impacts would occur in the males. What could be evaluated in males as a counterpart to pheromone production?

3.       Figure 1e is measuring fecundity, not fertility.

4.       Would weak, irradiated females that are co-released with semi-sterile males distract those males from seeking wild mates? Why would the males attempt to seek wild females when females are readily accessible?

5.       Do males/females seek multiple mates in this species. This could impact the efficacy of the proposed SIT/IS program.

Round 2

Reviewer 3 Report

Thank you to the authors for your responses. I am satisfied with the revisions and explanations.